# Evaluation of the efficacy of sulbactam combination therapy for monomicrobial and polymicrobial pulmonary infections caused by multidrug-resistant *Acinetobacter baumannii*

Liangfei Peng,[1,2] Xuemei Yang,[3] Xiaohua Qiu,[4] Li Wang,[5] Ling Chen,[1,6] Jiachen Wei,[1] Chuwei Jing,[1] Xiancheng Wu,[1,7] Wen Li,[1] Danni Wang,[1] Qian Qian,[8] Wenkui Sun[1]

**ABSTRACT**  The efficacy in treatment of monomicrobial versus polymicrobial *Acinetobacter baumannii* (mono-AB vs poly-AB) pulmonary infections remains unclear. This study mainly investigated the efficacy of sulbactam combination regimens against mono-AB and poly-AB pulmonary infections. A multicenter retrospective study of adult patients who received sulbactam or cefoperazone/sulbactam for multidrug-resistant *A. baumannii* (MDRAB) between August 2021 and July 2023 was conducted. The outcomes of 7-day microbiological efficacy and 14-day clinical efficacy were measured. A total of 366 patients were enrolled, including 75 mono-AB patients. Among the 291 poly-AB cases, 82 patients were co-infected with *Pseudomonas aeruginosa* and 76 patients with *Klebsiella pneumoniae*. All *A. baumannii* isolates exhibited multidrug resistance. The antibiotics demonstrating the highest sensitivity against these isolates were polymyxins and tigecycline, with sensitivity rates of 91.25% and 71.63%, respectively. In the multivariable analysis, patients receiving sulbactam-based treatment for poly-AB pulmonary infections exhibited significantly higher microbial clearance of *A. baumannii* compared with those with mono-AB infections. Microbial efficacy analysis of *A. baumannii* co-infection with *K. pneumoniae* or *P. aeruginosa* showed similar results to overall poly-AB infections. Sulbactam doses ≥ 8 g/day showed the best microbiological and clinical efficacy. Sulbactam combined with tigecycline and/or polymyxins demonstrated significantly better microbiological efficacy compared with sulbactam alone. Sulbactam combined with polymyxins resulted in better microbiological and clinical efficacy than when combined with tigecycline. In conclusion, patients with poly-AB lung infections exhibit a higher microbial clearance rate of *A. baumannii* following treatment compared with those with mono-AB infections. A daily dose ≥8 g sulbactam may be optimal for treating MDRAB pneumonia. The combination of sulbactam and polymyxins demonstrates significant advantages over other sulbactam-containing regimens in terms of microbiological and clinical efficacy.

**IMPORTANCE**  This is the first multicenter retrospective study to investigate the efficacy of sulbactam combination regimens in the treatment of monomicrobial and polymicrobial AB-related pulmonary infections. Our study found that patients receiving sulbactam-based treatment for poly-AB pulmonary infections exhibited significantly higher microbial clearance of *A. baumannii* compared with those with mono-AB infections. Microbial efficacy analysis of *A. baumannii* co-infection with *K. pneumoniae* or *P. aeruginosa* showed similar results to overall poly-AB infections. These findings are significantly important for further analyzing the potential mechanisms underlying the differences in efficacy between antimicrobial treatments for monomicrobial and polymicrobial infections, as well as for optimizing clinical antimicrobial combination strategies.

**Peer Reviewer** Hua Zhou, The First Affiliated Hospital of Zhejiang University School of Medicine, Hangzhou, China

Address correspondence to Wenkui Sun, sunwenkui@njmu.edu.cn, or Qian Qian, honeyhoney2007@163.com.

Liangfei Peng and Xuemei Yang contributed equally to this article. Author order was determined based on their contributions and seniority.

The authors declare no conflict of interest.

See the funding table on p. 12.

**KEYWORDS** *Acinetobacter baumannii*, polymicrobial pulmonary infection, multidrug-resistant, sulbactam, combination therapy, efficacy

*A*cinetobacter baumannii is a non-fermenting Gram-negative coccobacillus that acts as an opportunistic pathogen in nosocomial infections (1). It is typically associated with a higher incidence among patients undergoing mechanical ventilation in intensive care units (ICUs) (2). Respiratory infections are the most common type of infections caused by *A. baumannii*, especially those involving carbapenem-resistant *A. baumannii* (CRAB) strains (3). In recent years, the incidence and drug resistance of CRAB in clinical specimens have been steadily increasing (4). Reportedly, CRAB strains constitute 71.7% of all *A. baumannii* isolates in the Asia-Pacific region and exhibit high resistance against most antibiotics, except tigecycline and polymyxins (5). Global studies on multidrug-resistant *A. baumannii* (MDRAB) show that carbapenem resistance rates in the Organisation for Economic Cooperation and Development countries rose significantly from 23.8% in 2000 to 73.9% in 2016 (6). Infections caused by MDRAB are associated with high mortality rates, posing significant challenges to clinical treatment (7).

The remarkable ability of *A. baumannii* to develop antibiotic resistance limits available treatment options (8). The primary treatment regimens for MDRAB include polymyxins, tigecycline, sulbactam, and more recently cefiderocol, if available (9). No agent or combination regimen has demonstrated superiority over others in randomized clinical trials; nevertheless, sulbactam appears to have the best evidence for initial use (10). This benefit is likely attributed to its ability to saturate penicillin-binding proteins 1 and 3 when administered in high doses (10, 11). This ability highlights the unique intrinsic antibacterial activity of sulbactam against *A. baumannii* and supports the broader application of sulbactam at higher doses. Meanwhile, *A. baumannii* infections are often polymicrobial, complicating treatment efforts (12). Currently, research on the efficacy of sulbactam in treating monomicrobial AB (mono-AB) and polymicrobial AB (poly-AB)-related infections remains limited. Therefore, it is crucial to evaluate the clinical efficacy of sulbactam-based regimens in treating both mono-AB- and poly-AB-related lung infections. This evaluation will aid in understanding the potential mechanisms underlying the differences in efficacy between these two types of infections and in optimizing clinical antimicrobial drug combination regimens.

In this study, we collected electronic medical record data from multiple hospitals and retrospectively analyzed the efficacy of sulbactam in treating both mono-AB and poly-AB infections in patients with MDRAB pneumonia. To our knowledge, no similar study has been previously conducted. Furthermore, we investigated the clinical and microbiological efficacy of various sulbactam dosages and combination therapy regimens in these patients.

## MATERIALS AND METHODS

### Study design

In this multicenter retrospective study, data were collected from adult patients (age ≥18 years) with MDRAB pulmonary infection who received sulbactam treatment between August 2021 and July 2023. Further inclusion criteria required at least one sputum or bronchoalveolar lavage fluid (BALF) culture to be positive for MDRAB, and the sulbactam treatment duration was at least 3 days. Patients with human immunodeficiency virus infection, mental illness, a medication course of 2 days or less, or incomplete data were excluded. The collected data included demographic information, underlying diseases, mechanical ventilation status, duration of anti-infective treatment, Acute Physiology and Chronic Health Evaluation (APACHE) II score, presence of other infection sites and pathogens, sulbactam regimen, and concurrent medications.

## Etiological samples

Lower respiratory tract samples obtained with disposable sputum collectors, or collected deep sputum or BALF under fiberoptic bronchoscopy were collected from all patients for culture. Bacterial species were identified and tested for drug sensitivity using a VITEK 2 COMPACT (bioMérieux) automated microbiological identification system. The microbiology laboratories of the hospitals determined the minimum inhibitory concentrations (MICs) of sulbactam, polymyxins, and tigecycline using the agar dilution method. The drug sensitivity breakpoints for polymyxins were determined according to the 2020 standards of the European Committee on Antimicrobial Susceptibility Testing (EUCAST) (13). The determination basis of cefoperazone/sulbactam was the breakpoint standard of cefoperazone, and the situation of sulbactam was determined according to the drug sensitivity results of cefoperazone/sulbactam. The drug sensitivity breakpoints of tigecycline were cited from the standards of the U.S. Food and Drug Administration (FDA) (14).

## Related definitions

Pneumonia was defined as a new or progressive pulmonary infiltrate on chest radiography, along with at least two of the following characteristics: new onset fever >38°C or hypothermia <35.5°C, leukocytosis or leukopenia (leukocyte count >12,000 and <4,000 cells/mm$^3$, respectively), decline in oxygenation (O$_2$ saturation <90%), and increasing amount of purulent sputum (15). Multidrug-resistant (MDR) bacteria were considered to have acquired non-susceptibility to at least one agent in three or more antimicrobial categories (16). The pneumonia involving MDRAB was defined as the isolation of MDR *A. baumannii* in at least one sputum or BALF culture in a patient with temporally related clinical signs (17). The quality of tracheal aspirate and sputum specimens was considered adequate if Gram's stains revealed at least 25 neutrophils and fewer than 10 epithelial cells per low-power field. Bacterium growth was assessed semiquantitatively, and bacteria exhibiting at least moderate growth were defined as etiologic pathogens (18). Monomicrobial *A. baumannii* (mono-AB) was defined as only positive culture results for *A. baumannii* with time-related symptoms between 3 days before sample collection and the evaluation day (17). Polymicrobial *A. baumannii* (poly-AB) was defined as an infection with positive culture of *A. baumannii* with time-related symptoms and positive culture results of other bacteria, regardless of infection or colonization (19, 20).

## Study treatments

All patients were treated with sulbactam, including cefoperazone/sulbactam and/or sulbactam sodium for injection. The daily dose of sulbactam was determined by calculating the combined doses of cefoperazone/sulbactam and additional sulbactam sodium for injection. The selection, dosage, and duration of anti-MDRAB drugs were determined by the attending physician. For the analysis of clinical efficacy, only cases where the sulbactam treatment lasted 7 days or more were considered (17). Combination therapy was defined as the concurrent use of at least one *in vitro* coverage of anti-MDRAB drugs for at least 3 days. Anti-MDRAB treatments were analyzed according to two strategies. First, the treatments were categorized by the dose of sulbactam into ≤4, ≥6 to <8, and ≥8 g. Second, the treatments were categorized by the combination regimen of sulbactam into sulbactam-based therapy (excluding tigecycline and polymyxins), sulbactam plus tigecycline, sulbactam plus polymyxins, and sulbactam plus both tigecycline and polymyxins.

## Study assessments

The primary outcome measure was clinical improvement after 14 days of anti-MDRAB treatment, while the secondary outcome measure was microbiological improvement after 7 days of treatment. Clinical efficacy was classified as effective (characterized by the resolution or significant improvement of clinical symptoms and signs, normalization or

minimal residuals in pulmonary imaging, and normalization of inflammatory markers), uncertain or ineffective (21). Microbiological efficacy was defined as eradication or reduction, indicated by the absence of MDRAB in lower respiratory tract secretion cultures post-treatment, or a decrease in bacterial load by more than 25% compared with pre-treatment levels. Microbiological ineffectiveness was defined as the continued presence of MDRAB in cultures after treatment and no change or an increase in bacterial load compared with pre-treatment levels (22).

## Statistical analysis

All data were statistically analyzed using IBM SPSS 26.0 (IBM Corp). Quantitative data were presented as median and interquartile range (IQR). The *t*-test or Mann–Whitney *U*-test was used depending on whether the data conformed to a normal distribution. Qualitative data were expressed as numbers and percentages and compared between groups using Pearson's Chi-squared test and Fisher's exact test, as appropriate. Binary logistic regression and multivariate logistic regression were employed to analyze the 7-day microbial efficacy and 14-day clinical efficacy, respectively. In the multivariate analysis, all biologically plausible variables with $P < 0.20$ in the univariate analysis were included in the logistic regression model. Sulbactam dose and sulbactam combination regimens were analyzed separately to avoid duplicate variables in the multivariate analysis. Full factorial entry and main effects models were used to analyze microbial efficacy and clinical efficacy, respectively. The performance and suitability of the model were evaluated using the Hosmer-Lemeshow– goodness-of-fit test. Odds ratio (OR) and 95% confidence interval (CI) were reported, and bilateral $P < 0.05$ was considered statistically significant.

## RESULTS

A total of 75 mono-AB patients and 291 poly-AB patients were included in the 7-day microbiological efficacy analysis (Fig. 1). Among the poly-AB patients, 76 cases were co-infected with *Klebsiella pneumoniae*, and 82 cases with *Pseudomonas aeruginosa*. Moreover, 48 mono-AB patients and 233 poly-AB patients were evaluated for 14-day clinical efficacy, excluding patients treated with sulbactam fewer than 7 days. Finally, 54 and 65 poly-AB patients co-infected with *K. pneumoniae* and *P. aeruginosa,* respectively, remained.

## Characteristics of patients

A total of 366 patients with MDRAB pneumonia were enrolled for microbiological efficacy analysis, including 243 males (66.4%). The median age of these patients was 67 years (IQR 55, 76), and median APACHE II score was 19 (IQR 14, 22). Of these patients, 94.5% ($n = 346$) were admitted to ICU, and the median course of treatment was 10 days (IQR 6, 15). About 72.1% of patients ($n = 264$) had chronic underlying diseases, with hypertension being the most common (181 patients, 49.5%). Additionally, 40.2% of patients ($n = 147$) had infections other than pneumonia. Compared with the mono-AB group, the poly-AB group had longer courses of antimicrobial therapy and a higher proportion of infections other than pneumonia, particularly urinary tract infections.

In the analysis of clinical efficacy, 281 patients who received sulbactam for at least 7 days were included. The overall baseline characteristics of these patients were similar to those of the patients in the microbiological evaluation. The incidence of complicated infections other than pneumonia, especially urinary tract infections, was significantly different between the mono-AB and poly-AB groups, while other baseline characteristics showed no significant differences (Table 1).

## Treatment regimen

Among the 366 patients, 98 received additional single preparation of sulbactam (sulbactam sodium for injection). The most common therapeutic dose was ≤4 g (73.2%),

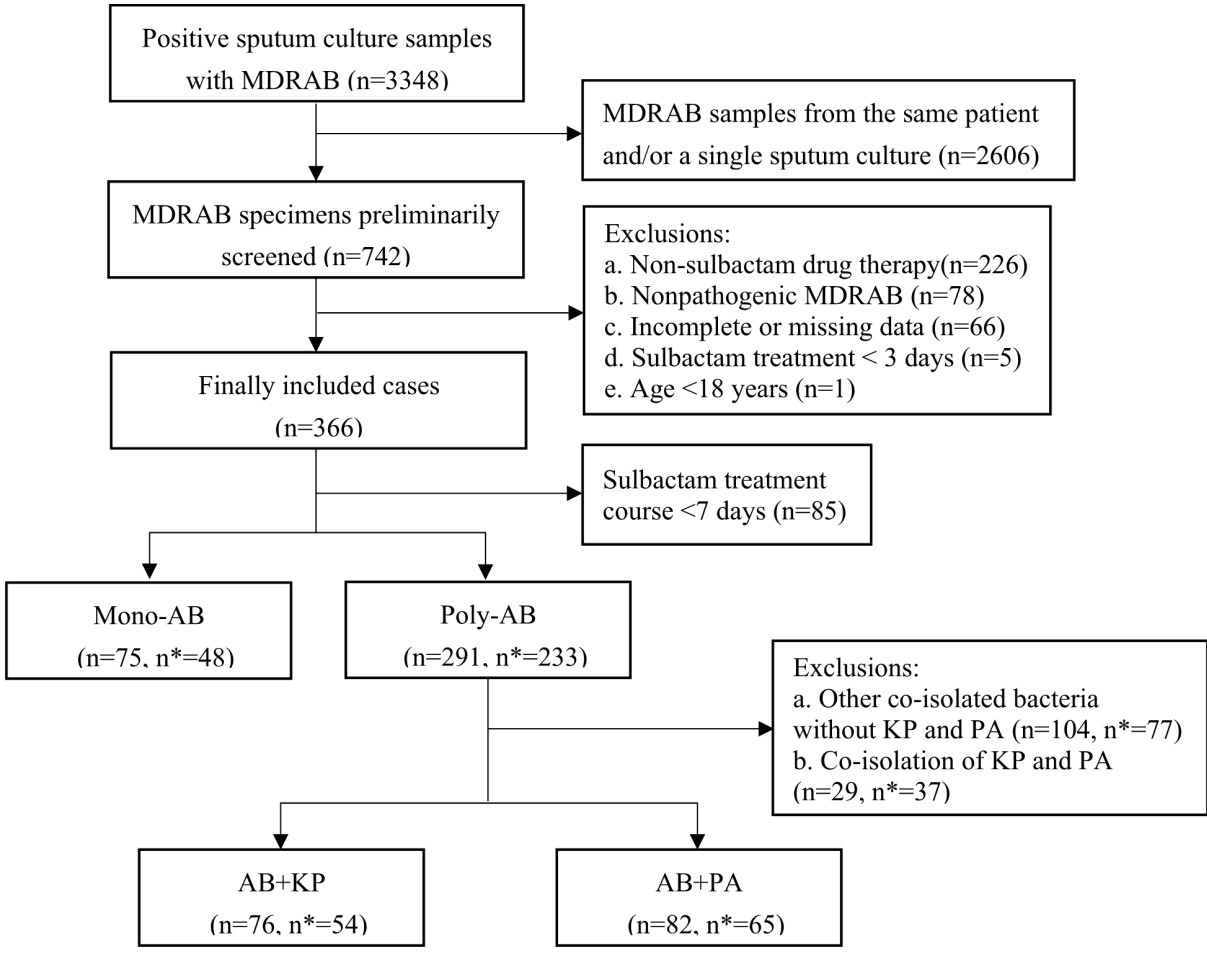

**FIG 1** Flowchart of patient enrollment. MDRAB, multidrug-resistant *A. baumannii*; Mono-AB, monomicrobial *A. baumannii*; Poly-AB, polymicrobial *A. baumannii*; AB, *A. baumannii*; KP, *K. pneumoniae*; PA, *P. aeruginosa*. * Indicates the number of cases analyzed for clinical efficacy.

while the numbers of patients receiving doses of ≥6 to <8 and ≥8 g were approximately equal. The predominant anti-MDRAB treatment strategies were sulbactam-based therapy and sulbactam plus tigecycline (41.8% and 37.7%, respectively), followed by sulbactam plus polymyxins and sulbactam plus tigecycline and polymyxins. The clinical efficacy data were similar (Table 1).

## Strain distribution in poly-AB patients

A total of 446 bacterial strains, excluding fungi and viruses, were co-isolated with *A. baumannii* in 366 patients diagnosed with MDRAB pneumonia (Fig. 2). *P. aeruginosa* and *K. pneumoniae* were the most frequently co-isolated species, accounting for 24.89% and 23.54% of cases, respectively. These species were followed by *Stenotrophomonas maltophilia*, *Burkholderia cepacia*, *Serratia marcescens*, *Staphylococcus species*, *Enterobacter cloacae*, *Escherichia coli*, and other *Klebsiella* species. Additionally, five cases (1.12%) involved other bacteria species, including *Raoultella* (n = 2), *Haemophilus influenzae* (n = 1), *Escherichia fergusonii* (n = 1), and *Pseudomonas putida* (n = 1).

## Susceptibility testing

All *A. baumannii* isolates examined here exhibited multidrug resistance. The susceptibility results for 19 antibiotics are presented in Fig. 3. The resistance rates of *A. baumannii* to β-lactamase inhibitors (excluding cefoperazone/sulbactam), third- and fourth-generation cephalosporins, carbapenems, fluoroquinolones, and trimethoprim/sulfamethoxazole

**TABLE 1** Characteristics of patients with monomicrobial and polymicrobial AB infections[g]

| Characteristic | Patients with microbiological efficacy | | | | Patients with clinical efficacy | | | |
|---|---|---|---|---|---|---|---|---|
| | All patients | Mono-AB | Poly-AB | $P^f$ | All patients | Mono-AB | Poly-AB | $P^f$ |
| | (n = 366) | (n = 75) | (n = 291) | | (n = 281) | (n = 48) | (n = 233) | |
| Age (year, median [IQR]) | 67 [55, 76] | 71 [59, 81] | 67 [55, 74] | 0.051 | 67 [56, 74] | 68 [57.5, 77.75] | 67 [55.5, 73.5] | 0.494 |
| Male | 243 (66.4) | 49 (65.3) | 194 (66.7) | 0.891 | 177 (63.0) | 29 (60.4) | 148 (63.5) | 0.743 |
| APACHE II score (median [IQR]) | 19 [14, 22] | 17 [13, 22] | 19 [15, 22] | 0.234 | 19 [14.5, 22] | 17 [12, 22.75] | 19 [15, 22] | 0.069 |
| Mechanical Ventilation | 295 (80.6) | 59 (78.7) | 236 (81.1) | 0.744 | 233 (82.9) | 36 (75.0) | 197 (84.5) | 0.139 |
| Admitted to ICU | 346 (94.5) | 71 (94.7) | 275 (94.5) | 1.000 | 270 (96.1) | 44 (91.7) | 226 (97.0) | 0.185 |
| Antimicrobial drug course (day, median [IQR]) | 10 [6, 15] | 8 [6, 13] | 10 [6, 15] | **0.042** | 12 [9, 17] | 12 [8.25, 16] | 12 [9, 17] | 0.365 |
| Chronic underlying diseases | 264 (72.1) | 60 (80.0) | 204 (70.1) | 0.112 | 206 (73.3) | 39 (81.3) | 167 (71.7) | 0.211 |
| Diabetes | 76 (20.8) | 14 (18.7) | 62 (21.3) | 0.638 | 63 (22.4) | 9 (18.8) | 54 (23.2) | 0.573 |
| Hypertension | 181 (49.5) | 44 (58.7) | 137 (47.1) | 0.092 | 142 (50.5) | 28 (58.3) | 114 (48.9) | 0.269 |
| Heart disease[a] | 97 (26.5) | 22 (29.3) | 75 (25.8) | 0.558 | 76 (27.0) | 14 (29.2) | 62 (26.6) | 0.723 |
| Cerebrovascular disease | 63 (17.2) | 16 (21.3) | 47 (16.2) | 0.305 | 38 (13.5) | 6 (12.5) | 32 (13.7) | 1.000 |
| Malignant tumor | 29 (7.9) | 5 (6.7) | 24 (8.2) | 0.812 | 25 (8.9) | 5 (10.4) | 20 (8.6) | 0.898 |
| Complicated with infection other than pneumonia[b] | 147 (40.2) | 19 (25.3) | 128 (44.0) | **0.004** | 121 (43.1) | 10 (20.8) | 111 (47.6) | **0.001** |
| Bloodstream infection | 61 (16.7) | 7 (9.3) | 54 (18.6) | 0.081 | 57 (20.3) | 5 (10.4) | 52 (22.3) | 0.075 |
| Urinary tract infection | 55 (15.0) | 4 (5.3) | 51 (17.5) | **0.010** | 43 (15.3) | 2 (4.2) | 41 (17.6) | **0.025** |
| Intracranial infection | 17 (4.6) | 3 (4.0) | 14 (4.8) | 1.000 | 8 (2.8) | 1 (2.1) | 7 (3.0) | 1.000 |
| Skin and soft tissue infection | 15 (4.1) | 0 (0.0%) | 15 (5.2) | 0.093 | 15 (5.3) | 0 (0.0) | 15 (6.4) | 0.146 |
| Sulbactam daily dosage | | | | | | | | |
| ≤4 g (ref) | 268 (73.2) | 58 (77.3) | 210 (72.2) | 0.657 | 192 (68.3) | 38 (79.2) | 154 (66.1) | 0.122 |
| ≥6–< 8g[c] | 48 (13.1) | 8 (10.7) | 40 (13.7) | | 46 (16.4) | 7 (14.6) | 39 (16.7) | |
| ≥8 g | 50 (13.7) | 9 (12.0) | 41 (14.1) | | 43 (15.3) | 3 (6.3) | 40 (17.2) | |
| Sulbactam combination regimens | | | | | | | | |
| Sulbactam-based[d] (ref) | 153 (41.8) | 41 (54.7) | 112 (38.5) | **0.048** | 92 (32.7) | 20 (41.7) | 72 (30.9) | 0.519 |
| Sulbactam + tigecycline | 138 (37.7) | 20 (26.7) | 118 (40.5) | | 109 (38.8) | 17 (35.4) | 92 (39.5) | |
| Sulbactam + polymyxins[e] | 39 (10.7) | 9 (12.0) | 30 (10.3) | | 32 (11.4) | 4 (8.3) | 28 (12.0) | |
| Sulbactam + polymyxins[e] + tigecycline | 36 (9.8) | 5 (6.7) | 31 (10.7) | | 48 (17.1) | 7 (14.6) | 41 (17.6) | |

[a]Heart disease encompasses coronary heart disease, arrhythmias, and heart failure.
[b]Complication with infection other than pneumonia was defined as infections identified within 3 days before and after pneumonia.
[c]Sulbactam dosage was not found using 5 g/day.
[d]This refers to sulbactam alone or in combination with other antibiotics, excluding tigecycline and polymyxins.
[e]Polymyxins refer to either polymyxin B or polymyxin E (colistin).
[f]P values were calculated by t-test, Mann–Whitney U-test, or chi-square test as appropriate.
[g]Data are presented as median (25th–75th percentiles) or N (%). Statistically significant P values are highlighted in bold. APACHE, Acute Physiology and Chronic Health Evaluation; ICU, intensive care unit.

(SXT) all exceeded 90%. The resistance rates to cefoperazone/sulbactam (CSL), amikacin (AN), and minocycline (MNO) were moderate, ranging from 22.78% to 56.43%. In contrast, the resistance rates to polymyxins and tigecycline (TGC) were the lowest (1.25% and 4.96%, respectively). *K. pneumoniae* and *P. aeruginosa* co-isolated with *A. baumannii* exhibited relatively lower resistance to the 19 antibiotics compared with *A. baumannii* (Fig. S1 and S2).

## Comparison of efficacy in mono-AB and poly-AB

Differences in microbial efficacy against *A. baumannii* were observed between patients with mono-AB infections and those with poly-AB-related lung infections (Table 2). In the multivariable analysis, patients with poly-AB infections exhibited significantly higher microbiological efficacy against AB post-treatment compared with those with mono-AB infections (adjusted odds ratio [aOR] 2.073, 95% CI 1.226–3.506, P = 0.007). Specifically, patients with *A. baumannii* co-infected with *K. pneumoniae* (AB + KP) or *P. aeruginosa* (AB

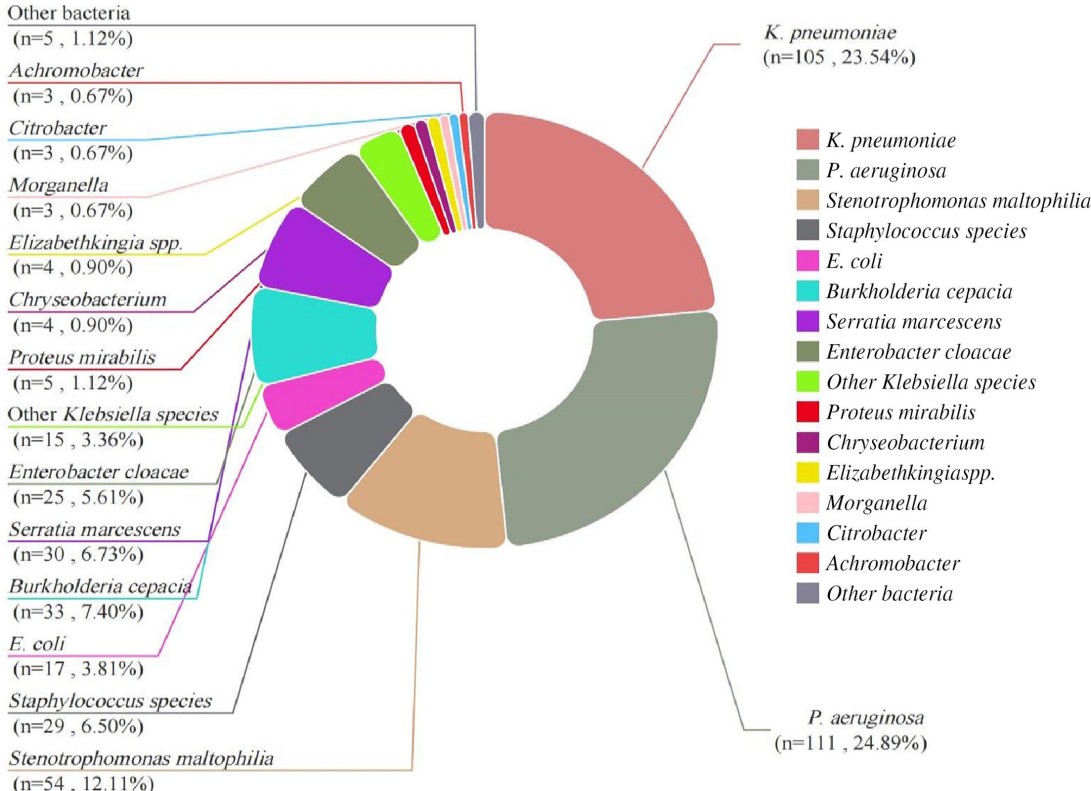

**Other bacteria**
(n=5 , 1.12%)

*Achromobacter*
(n=3 , 0.67%)

*Citrobacter*
(n=3 , 0.67%)

*Morganella*
(n=3 , 0.67%)

*Elizabethkingia spp.*
(n=4 , 0.90%)

*Chryseobacterium*
(n=4 , 0.90%)

*Proteus mirabilis*
(n=5 , 1.12%)

Other *Klebsiella* species
(n=15 , 3.36%)

*Enterobacter cloacae*
(n=25 , 5.61%)

*Serratia marcescens*
(n=30 , 6.73%)

*Burkholderia cepacia*
(n=33 , 7.40%)

*E. coli*
(n=17 , 3.81%)

*Staphylococcus species*
(n=29 , 6.50%)

*Stenotrophomonas maltophilia*
(n=54 , 12.11%)

*K. pneumoniae*
(n=105 , 23.54%)

*P. aeruginosa*
(n=111 , 24.89%)

Legend:
- *K. pneumoniae*
- *P. aeruginosa*
- *Stenotrophomonas maltophilia*
- *Staphylococcus species*
- *E. coli*
- *Burkholderia cepacia*
- *Serratia marcescens*
- *Enterobacter cloacae*
- *Other Klebsiella species*
- *Proteus mirabilis*
- *Chryseobacterium*
- *Elizabethkingiaspp.*
- *Morganella*
- *Citrobacter*
- *Achromobacter*
- *Other bacteria*

**FIG 2** Distribution of bacterial species in patients with poly-AB.

+ PA) demonstrated significantly improved microbiological efficacy against *A. baumannii* post-treatment compared with patients with mono-AB infections ($P$ = 0.035 and $P$ = 0.025, respectively). However, within the poly-AB group, the post-treatment microbiological efficacy between AB + KP and AB + PA infections is comparable (aOR 1.051, 95% CI 0.528–2.091, $P$ = 0.888).

In the concurrent evaluation of clinical efficacy, patients with AB + KP or AB + PA infections as well as those with poly-AB infections did not significantly differ from patients with mono-AB infections ($P$ > 0.05). Similarly, there was no significant difference between patients with AB + KP and those with AB + PA infections (aOR 0.694, 95% CI 0.271–1.780, $P$ = 0.448) (Table 2).

## Evaluation of sulbactam efficacy at varying doses

Sulbactam had different therapeutic effects on the patients with MDRAB pneumonia depending on the daily dose (Table 3). In the multivariable analysis, the microbiological efficacy of sulbactam at dose ≥6 g was superior over that of dose ≤4 g (aOR 2.162, 95% CI 1.216–3.844, $P$ = 0.009), whereas the clinical efficacy of the two dosage groups was comparable ($P$ = 0.286). Specifically, a sulbactam dose ≥8 g significantly enhanced microbiological efficacy over a dose ≤4 g (aOR 2.438, 95% CI 1.124–5.291, $P$ = 0.024), with improved clinical efficacy as well ($P$ = 0.015). The clinical efficacy of dose ≥8 g surpassed that of 6 g ≤ dose < 8 g (aOR 5.170, 95% CI 1.512–17.685, $P$ = 0.009), while microbiological efficacy was comparable between these two groups ($P$ = 0.640). Furthermore, no significant differences were observed in either microbiological or clinical efficacy between 6 g ≤ dose < 8 g and ≤4 g ($P$ = 0.078 and $P$ = 0.464, respectively). Tables S1 and S2 detailed the key variables associated with microbiological and clinical efficacy, respectively.

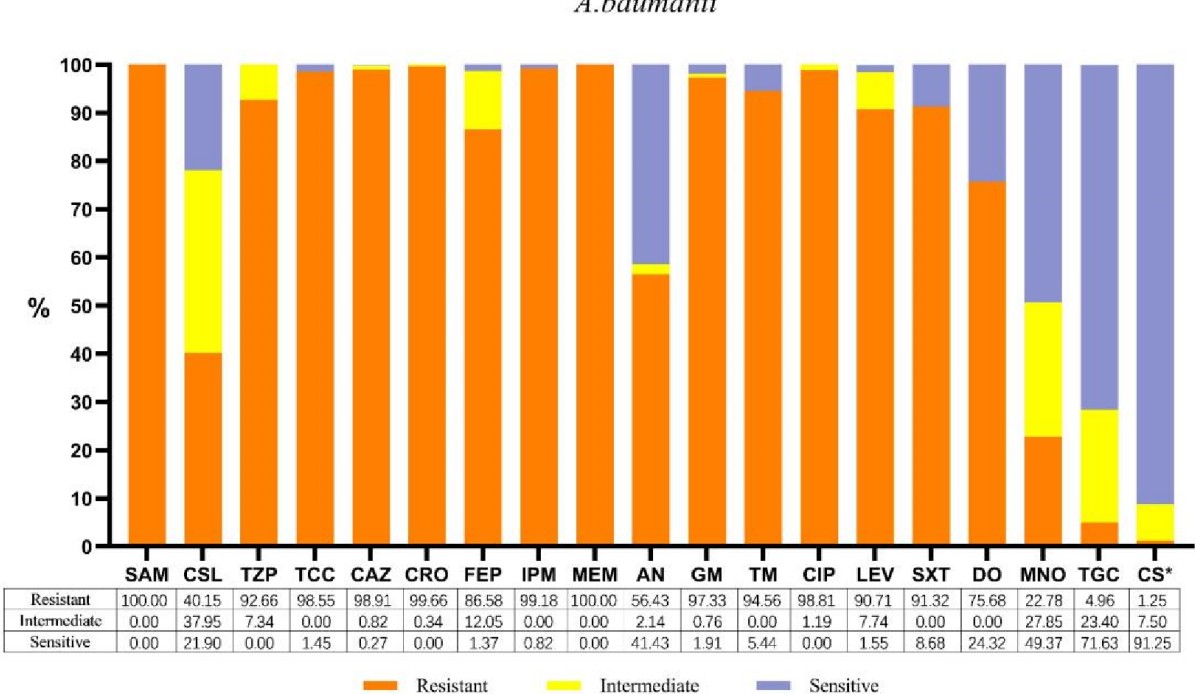

**FIG 3** Antimicrobial sensitivity patterns of *A. baumannii* clinical isolates against 19 antibiotics. The abbreviations, as they appear in the figure, are as follows: SAM, ampicillin/sulbactam (2/1); CSL, cefoperazone/sulbactam (2/1); TZP, piperacillin/tazobactam; TCC, ticarcillin/clavulanic acid; CAZ, ceftazidime; CRO, ceftriaxone; FEP, cefepime; IPM, imipenem; MEM, meropenem; AN, amikacin; GM, gentamicin; TM, tobramycin; CIP, ciprofloxacin; LEV, levofloxacin; SXT, trimethoprim/sulfamethoxazole; DO, doxycycline; MNO, minocycline; TGC, tigecycline; CS, colistin. * The sensitivity results for colistin and polymyxin B are equivalent.

## Efficacy evaluation of different sulbactam combination regimens

The efficacy of different sulbactam combination regimens varied in treating patients with MDRAB pneumonia (Table 4). In the multivariable analysis, the microbial efficacy of sulbactam combined with tigecycline and/or polymyxins is significantly superior over sulbactam monotherapy ($P < 0.001$). Sulbactam plus polymyxins demonstrates higher microbiological efficacy compared with sulbactam plus tigecycline (aOR 2.850, 95% CI 1.023–7.937, $P = 0.045$). The microbiological efficacy of the sulbactam plus tigecycline regimen was significantly enhanced after the addition of polymyxins (aOR 4.733, 95% CI 1.347–16.628, $P = 0.015$).

Moreover, the clinical efficacy of sulbactam plus polymyxin regimen surpasses that of sulbactam monotherapy (aOR 3.672, 95% CI 1.040–12.968, $P = 0.043$), despite the absence of statistical significance in univariate analysis. Additionally, the clinical efficacy of sulbactam combined with polymyxins exceeds that of sulbactam combined with tigecycline (aOR 5.173, 95% CI 1.471–18.191, $P = 0.010$). No significant differences were observed in the clinical efficacy of other groups (Table 4). Baseline variables associated with microbiological and clinical efficacy are shown in Tables S1 and S2.

## DISCUSSION

To our knowledge, this is the first multicenter study to compare sulbactam-based treatment regimens and assess their efficacy in treating both mono-AB- and poly-AB-related lung infections. Overall, sulbactam-based regimens demonstrated greater microbiological efficacy against *A. baumannii* in poly-AB-related lung infections compared with mono-AB infections. In poly-AB patients co-infected with *K. pneumoniae* or *P. aeruginosa*, sulbactam-containing treatments demonstrated superior

**TABLE 2** Differences in efficacy between monomicrobial and polymicrobial AB lung infections[c]

| Type of AB infection | 7-day microbiological efficacy | | | | 14-day clinical efficacy | | | |
|---|---|---|---|---|---|---|---|---|
| | Effective/total (n, %) | P | Adjusted OR (95% CI)[a] | P | Effective/total (n, %) | P | Adjusted OR (95% CI)[b] | P |
| Mono-AB vs poly-AB[a] | | | | | | | | |
| Mono-AB (ref) | 36/75 (48.0) | | (ref） | | 18/48 (37.5) | | (ref） | |
| Poly-AB | 199/291 (68.4) | **0.001** | 2.073 (1.226–3.506) | **0.007** | 90/233 (38.6) | 0.467 | 0.987 (0.413–2.356) | 0.976 |
| Mono-AB vs AB + KP[a] | | | | | | | | |
| Mono-AB (ref) | 36/75 (48.0) | | (ref） | | 18/48 (37.5) | | (ref） | |
| AB + KP | 51/76 (67.1) | **0.021** | 2.046 (1.051–3.985) | **0.035** | 26/54 (48.1) | 0.330 | 1.353 (0.476–3.849) | 0.570 |
| Mono-AB vs. AB + PA[b] | | | | | | | | |
| Mono-AB (ref) | 36/75 (48.0) | | (ref） | | 18/48 (37.5) | | (ref） | |
| AB + PA | 58/82 (70.7) | **0.005** | 2.159 (1.100–4.237) | **0.025** | 23/65 (35.4) | 0.806 | 0.928 (0.331–2.600) | 0.887 |
| AB + KP vs AB + PA | | | | | | | | |
| AB + KP (ref) | 51/76 (67.1) | | (ref） | | 26/54 (48.1) | | (ref） | |
| AB + PA | 58/82 (70.7) | 0.731 | 1.051 (0.528–2.091) | 0.888 | 23/65 (35.4) | 0.330 | 0.694 (0.271–1.780) | 0.448 |

[a]In the multivariate analysis of microbiological efficacy, the factors including antimicrobial drug course, bloodstream infection, and skin and soft tissue infection were adjusted to calculate the OR and 95% CI.
[b]In the multivariate analysis of clinical efficacy, factors such as APACHE II score and urinary tract infection were adjusted to calculate the OR and 95% CI.
[c]Data are presented as N (%). Statistically significant P values are highlighted in bold. AB, *Acinetobacter baumannii*; KP, *Klebsiella pneumoniae*; PA, *Pseudomonas aeruginosa*; OR, odds ratio; CI, confidence interval.

microbiological efficacy against *A. baumannii* compared with treatments for mono-AB patients. The study results indicate that for poly-AB-related lung infections, anti-AB treatments are more effective in eliminating *A. baumannii* than in mono-AB infections. The topic of monomicrobial versus polymicrobial infections has gained increasing clinical attention. Research on mixed enterococcal bloodstream infections indicates patients with mixed infections generally have worse prognosis than those with monomicrobial infections (23). Similarly, mixed Candida/bacterial bloodstream infections and co-infections with carbapenem-resistant *Enterobacterales* (CRE) and *P. aeruginosa* also have higher mortality rates, emphasizing the complexity of managing polymicrobial infections (24, 25). The primary pathogens in these studies are not *A. baumannii*, and the infections are predominantly bloodstream infections, with research outcomes focusing on clinical prognosis. Currently, there is a paucity of research on the efficacy of treatments for mono-AB versus poly-AB-related lung infections.

**TABLE 3** Comparison of efficacy among different doses of sulbactam[c]

| Sulbactam daily dosage | 7-day microbiological efficacy | | | | 14-day clinical efficacy | | | |
|---|---|---|---|---|---|---|---|---|
| | Effective/total (n, %) | P | Adjusted OR (95% CI)[a] | P | Effective/total (n, %) | P | Adjusted OR (95% CI)[b] | P |
| ≤4 g vs. ≥6 g | | | | | | | | |
| ≤4 g (ref) | 159/268 (59.3) | | (ref) | | 72/192 (37.5) | | (ref) | |
| ≥6 g | 76/98 (77.6) | **0.001** | 2.162 (1.216–3.844) | **0.009** | 36/89 (40.4) | 0.552 | 1.448 (0.734–2.857) | 0.286 |
| ≤4 g vs. ≥6–< 8 g | | | | | | | | |
| ≤4 g (ref) | 159/268 (59.3) | | (ref) | | 72/192 (37.5) | | (ref) | |
| ≥6–< 8 g | 36/48 (75.0) | 0.052 | 1.933 (0.929–4.025) | 0.078 | 14/46 (30.4) | 0.665 | 0.734 (0.320–1.680) | 0.464 |
| ≤4 g vs. ≥8 g | | | | | | | | |
| ≤4 g (ref) | 159/268 (59.3) | | (ref) | | 72/192 (37.5) | | (ref) | |
| ≥8 g | 40/50 (80.0) | **0.006** | 2.438 (1.124–5.291) | **0.024** | 22/43 (51.2) | 0.057 | 3.793 (1.300–11.066) | **0.015** |
| ≥6–< 8 g vs. ≥8 g | | | | | | | | |
| ≥6–< 8 g (ref) | 36/48 (75.0) | | (ref) | | 14/46 (30.4) | | (ref) | |
| ≥8 g | 40/50 (80.0) | 0.632 | 1.261 (0.477–3.332) | 0.640 | 22/43 (51.2) | **0.035** | 5.170 (1.512–17.685) | **0.009** |

[a]In the multivariate analysis of microbiological efficacy, the factors including APACHE II score, mechanical ventilation, bloodstream infection, and *P. aeruginosa* were adjusted to calculate the OR and 95% CI.
[b]In the multivariate analysis of clinical efficacy, factors such as age and urinary tract infection were adjusted to calculate the OR and 95% CI.
[c]Data are presented as N (%). Statistically significant P values are highlighted in bold.

**TABLE 4** Comparison of efficacy of different sulbactam combination regimens[d]

| Sulbactam regimen | 7-day microbiological efficacy | | | | 14-day clinical efficacy | | | |
|---|---|---|---|---|---|---|---|---|
| | Effective/total (n, %) | P | Adjusted OR (95% CI)[b] | P | Effective/total (n, %) | P | Adjusted OR (95% CI)[c] | P |
| SUL vs. SUL + TGC + POL | | | | | | | | |
| SUL-based[a] (ref) | 69/153(45.1) | | (ref) | | 41/92(44.6) | | (ref) | |
| SUL +TGC + POL | 33/36 (91.7) | < 0.001 | 12.256 (3.478–43.183) | < 0.001 | 17/48(35.4) | 0.073 | 1.320 (0.490–3.554) | 0.583 |
| TGC vs. POL | | | | | | | | |
| SUL +TGC (ref) | 99/138 (71.7) | | (ref) | | 33/109(30.3) | | (ref) | |
| SUL +POL | 34/39 (87.2) | 0.059 | 2.850 (1.023–7.937) | **0.045** | 17/32(53.1) | **0.034** | 5.173 (1.471–18.191) | **0.010** |
| TGC vs. non-TGC | | | | | | | | |
| SUL-based[a] (ref) | 69/153 (45.1) | | (ref) | | 41/92(44.6) | | (ref) | |
| SUL + TGC | 99/138 (71.7) | < 0.001 | 2.589 (1.532–4.377) | < 0.001 | 33/109(30.3) | 0.051 | 0.710 (0.348–1.447) | 0.346 |
| SUL + POL (ref) | 34/39 (87.2) | | (ref) | | 17/32(53.1) | | (ref) | |
| SUL + POL + TGC | 33/36 (91.7) | 0.799 | 1.661 (0.361–7.638) | 0.515 | 17/48(35.4) | 0.272 | 0.359 (0.088–1.470) | 0.155 |
| POL vs. non-POL | | | | | | | | |
| SUL-based[a] (ref) | 69/153 (45.1) | | (ref) | | 41/92(44.6) | | (ref) | |
| SUL + POL | 34/39 (87.2) | < 0.001 | 7.379 (2.650–20.545) | < 0.001 | 17/32(53.1) | 0.129 | 3.672 (1.040–12.968) | **0.043** |
| SUL + TGC (ref) | 99/138 (71.7) | | (ref) | | 33/109(30.3) | | (ref) | |
| SUL + TGC + POL | 33/36 (91.7) | **0.015** | 4.733 (1.347–16.628) | **0.015** | 17/48(35.4) | 0.469 | 1.860 (0.696–4.972) | 0.216 |

[a]Refers to sulbactam alone or in combination with other antibiotics, excluding tigecycline and polymyxins.
[b]In the multivariate analysis of microbiological efficacy, factors such as APACHEII score, mechanical ventilation, antibiotic course, heart disease, bloodstream infection and concurrent other pathogens were adjusted to calculate the OR and 95% CI.
[c]In the multivariate analysis of clinical efficacy, factors such as age, APACHEII score, heart disease and urinary tract infection were adjusted to calculate the OR and 95%CI.
[d]Data are presented as N (%). SUL, sulbactam; TGC, tigecycline; POL, polymyxin B or polymyxin E. Statistically significant P values are highlighted in bold.

*A. baumannii* infections are often polymicrobial, as a meta-analysis shows 27% of lung or bloodstream infections caused by *A. baumannii* involve multiple microorganisms (26). Co-infections with CRAB and carbapenem-resistant *K. pneumoniae* lead to higher morbidity, increased ICU admissions, and longer hospital stay (27). Additionally, patients co-infected with MDRAB and CRE have a 90-day all-cause mortality rate up to 33.5% (25). However, some studies suggest polymicrobial infections involving *A. baumannii* may have a lower 28-day mortality rate compared with infections caused by *A. baumannii* alone, which are possibly due to varying microbial interactions, such as antagonistic or synergistic effects (12, 28). Therefore, the variation in mortality rates highlights the importance of identifying other isolated pathogens alongside *A. baumannii*. A study on 290 cases of poly-AB pulmonary infections shows the common co-isolated bacteria are *P. aeruginosa* (36%), *Staphylococcus aureus* (28%), and *Klebsiella species* (11%) (26). In the present study, 79.5% of patients had mixed *A. baumannii* infections, possibly due to the inclusion of non-pathogenic co-isolates. The co-isolation rates of *P. aeruginosa* (28.2%) and *K. pneumoniae* (26.1%) were similar to previous findings. In comparison, *S. aureus* appeared less frequently (10%), which is potentially due to its presence in the nose, throat, or skin wounds (29). The frequent co-isolation of *A. baumannii* with these pathogens may suggest possible beneficial interactions.

In polymicrobial infections, bacteria often engage in mutually beneficial interactions through the release of enzymes or other mechanisms. As reported, carbapenemase produced by *A. baumannii* can protect carbapenem-sensitive *Enterobacteriaceae* or *P. aeruginosa* (30). Under *in vitro* co-culture conditions, *A. baumannii* can systematically increase the concentration of *S. aureus* and significantly reduce the efficacy of β-lactam antibacterial drugs against it (31). Additionally, cross-protection and cross-feeding between *A. baumannii* and *K. pneumoniae* may underpin their co-existence in polymicrobial infections (32). These synergistic interactions between *A. baumannii* and other pathogens can cause severe damage to the host. To disrupt these symbiotic interactions in poly-AB infections, it may be crucial to prioritize the treatment of *A. baumannii* and effectively reduce its pathogen load. Sulbactam is widely used to treat *A. baumannii* infections, owing to its ability to inhibit penicillin-binding proteins

(11) and its intrinsic antibacterial activity against *A. baumannii*. Our findings indicate that sulbactam may interfere with the activity of *A. baumannii* and other pathogens in polymicrobial infections, enhancing drug susceptibility. However, research on the mechanisms underlying the interactions of sulbactam with microorganisms in the context of polymicrobial infections remains limited, necessitating further investigation.

Significant differences exist in the efficacy of different sulbactam dosages against *A. baumannii* infections. Higher doses of sulbactam (≥6 g/day) reportedly are more effective than standard doses (≤4 g/day) (33, 34). The present study confirms these findings and demonstrates that sulbactam doses ≥ 6 g/day offer superior microbiological efficacy, while doses ≥ 8 g/day further enhance both microbiological and clinical efficacy. These results align with the recommendation from the Infectious Diseases Society of America for high-dose sulbactam (9 g/day) in treating moderate to severe CRAB infections (35). The optimal therapeutic dose of sulbactam is continuously refined in clinical practice. A randomized trial revealed that although microbiological efficacy was greater at 12 g/day on day 7, mortality rates were similar between the 12 g/day and 9 g/day groups at later stages (36). Our study found that sulbactam doses ≥ 8 g/day exhibited significantly improved clinical efficacy compared with doses of 6 to 8 g and showed a higher microbiological efficacy but not significantly (80% vs. 75%). These results suggest that a high-dose sulbactam regimen (≥8 g/day) may be optimal for the treatment of MDRAB pneumonia.

The combination of sulbactam with either tigecycline or polymyxins offers a promising therapeutic option for treating *A. baumannii* infections, especially when sulbactam alone is ineffective. Elsayed et al. demonstrated that a regimen of ampicillin-sulbactam combined with tigecycline significantly enhanced the microbial clearance rate of *A. baumannii* by day 14 (37). In consistency with these findings, our study confirms that the sulbactam-tigecycline combination enhances microbiological efficacy against *A. baumannii*, although it does not significantly improve clinical efficacy. A meta-analysis supporting this observation shows no substantial difference in clinical efficacy between tigecycline and non-tigecycline regimens in treating MDRAB infections (38). This lack of clinical benefit may be attributed to the low tigecycline concentration in epithelial lining fluids and alveolar cells (39). Additionally, low albumin levels might impair tigecycline efficacy, owing to the high plasma protein-binding rate of albumin (40). In contrast, another meta-analysis reveals the superior microbiological efficacy of colistin-based therapies over non-colistin regimens against *A. baumannii* (41). Our study found that a sulbactam-based regimen combined with polymyxins significantly improved microbiological and clinical efficacy compared with a sulbactam-based regimen. Specifically, the sulbactam-polymyxin regimen outperformed sulbactam-based therapy in both microbiological and clinical efficacy, whereas the sulbactam-tigecycline combination, despite higher microbiological efficacy, did not provide a significant clinical benefit over sulbactam-based therapy.

In the context of sulbactam-based combination therapies, assessing the efficacy of polymyxins and tigecycline against MDRAB infections remains a crucial topic in clinical research. While tigecycline-based and colistin-based treatments show comparable microbiological and clinical efficacy (42), colistin is notably more effective against extensively drug-resistant *A. baumannii* at the microbiological level (43). A systematic review indicates the combination of polymyxins and sulbactam is more effective than the combination of polymyxins and tigecycline, particularly in terms of microbiological efficacy (44). Our study found that a triple regimen comprising sulbactam, tigecycline, and polymyxins achieved the highest microbiological efficacy. However, this regimen did not present a significant advantage over the sulbactam-polymyxin combination. Consequently, the sulbactam and polymyxins combination emerges as the preferred treatment for MDRAB pneumonia and offers superior clinical and microbiological efficacy over other sulbactam combinations.

This study has several limitations. First, it was retrospective and non-randomized, and the variations in the type, amount, and duration of antibiotics between groups

potentially introduced bias. Although we statistically used a multivariate analysis model to adjust for confounding factors between different groups, there may still be some unknown confounding effects and risks of bias. Second, the study exclusively involved patients using sulbactam, lacking a comparison with non-sulbactam treatments. Third, the use of polymyxins was relatively limited, and the modes of administration (inhaled or intravenous) were not clearly differentiated. Future research employing prospective or matched-pair design is needed to evaluate the efficacy of inhaled versus intravenous polymyxins in treating MDRAB infections.

In conclusion, treatments for poly-AB-related pulmonary infections demonstrate superior microbiological efficacy against *A. baumannii* compared with mono-AB infections. Experiments on the efficacy of various sulbactam doses demonstrate a dose ≥8 g is optimal for treating MDRAB pneumonia, in terms of both microbiological and clinical efficacy. Furthermore, the combination of sulbactam with polymyxins exhibits significant advantages in both microbiological and clinical efficacy over other sulbactam treatment regimens. Future research is required to elucidate the mechanisms by which sulbactam facilitates the clearance of *A. baumannii* in polymicrobial infections.

## ACKNOWLEDGMENTS

This research was funded by the Major Project of Jiangsu Health Vocational College (No. JKA2021002) and the Open Project of Jiangsu Health Development Research Center (No. JSHD2022048). Additionally, support was provided by the 2024 Qing Lan Project of Jiangsu Province.

Conception and design of the study: W.S. and Q.Q. Data generation: L.P., X.Y., L.C., J.W., C.J., X.W., W.L., and D.W. Analysis and interpretation of the data: L.P., X.Y., and W.S. Preparation or critical revision of the manuscript: L.P. and W.S. Each certifies no conflicts of interest.

## AUTHOR AFFILIATIONS

[1]Department of Respiratory and Critical Care Medicine, The First Affiliated Hospital With Nanjing Medical University, Nanjing, Jiangsu, China

[2]Department of Respiratory and Critical Care Medicine, The Second Affiliated Hospital of Wannan Medical College, Wuhu, Anhui, China

[3]Department of Emergency and Critical Care Medicine, The First Affiliated Hospital of Wannan Medical College, Wuhu, Anhui, China

[4]Department of Respiratory and Critical Care Medicine, Nanjing Drum Tower Hospital, Affiliated Hospital of Nanjing University Medical School, Nanjing, Jiangsu, China

[5]Department of Respiratory and Critical Care Medicine, The Second Affiliated Hospital of Nanjing University of Chinese Medicine, Nanjing, Jiangsu, China

[6]Department of Respiratory and Critical Care Medicine, BenQ Medical Center, Affiliated BenQ Hospital of Nanjing Medical University, Nanjing, Jiangsu, China

[7]Jianhu Clinical College, Jiangsu Medical Vocational College, Yancheng, Jiangsu, China

[8]Jiangsu Health Vocational College, Nanjing, Jiangsu, China

## AUTHOR ORCIDs

Qian Qian http://orcid.org/0009-0007-7670-920X
Wenkui Sun http://orcid.org/0000-0002-2992-9783

## FUNDING

| Funder | Grant(s) | Author(s) |
|---|---|---|
| The Major Project of Jiangsu Health Vocational College | JKA2021002 | Qian Qian |
| The Open Project of Jiangsu Health Development Research Center | JSHD2022048 | Qian Qian |

## AUTHOR CONTRIBUTIONS

Liangfei Peng, Data curation, Formal analysis, Investigation, Writing – original draft, Writing – review and editing | Xuemei Yang, Data curation, Investigation | Xiaohua Qiu, Data curation, Investigation | Li Wang, Data curation, Investigation | Ling Chen, Data curation, Investigation | Jiachen Wei, Data curation, Investigation | Chuwei Jing, Investigation | Xiancheng Wu, Investigation | Wen Li, Data curation, Investigation | Danni Wang, Data curation, Investigation | Qian Qian, Funding acquisition, Resources | Wenkui Sun, Project administration, Supervision, Writing – review and editing

## ETHICS APPROVAL

This study was approved by the Institutional Review Committee of the First Affiliated Hospital with Nanjing Medical University (approval no. 2023-SR-138). All methods were carried out in accordance with relevant guidelines and regulations.

## ADDITIONAL FILES

The following material is available online.

### Supplemental Material

**Supplemental figures (Spectrum03355-24-s0001.pdf).** Fig. S1 and S2.
**Supplemental tables (Spectrum03355-24-s0002.pdf).** Tables S1 to S3.

### Open Peer Review

**PEER REVIEW HISTORY (review-history.pdf).** An accounting of the reviewer comments and feedback.

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
