## [Reviewer comments · Microbiology Spectrum]

Microbiology Spectrum

Evaluation of the Efficacy of Sulbactam Combination Therapy for Monomicrobial and Polymicrobial Pulmonary Infections Caused by Multidrug-Resistant *Acinetobacter baumannii*

Liangfei Peng, Xuemei Yang, Xiaohua Qiu, Li Wang, Ling Chen, Jiachen Wei, Chuwei Jing, Xiancheng Wu, Wen Li, Danni Wang, qian qian, and wenkui sun

Corresponding Author(s): wenkui sun, The First Affiliated Hospital With Nanjing Medical University

Review Timeline:

Submission Date:	December 20, 2024
Editorial Decision:	February 18, 2025
Revision Received:	March 21, 2025
Accepted:	March 26, 2025

Editor: Tulip Jhaveri

Reviewer(s): Disclosure of reviewer identity is with reference to reviewer comments included in decision letter(s). The following individuals involved in review of your submission have agreed to reveal their identity: Hua Zhou (Reviewer #1)

Transaction Report:

DOI: <https://doi.org/10.1128/spectrum.03355-24>

Re: Spectrum03355-24 (**Evaluation of the Efficacy of Sulbactam Combination Therapy for Monomicrobial and Polymicrobial Pulmonary Infections Caused by Multidrug-Resistant *Acinetobacter baumannii***)

Dear Dr. wenkui sun:

Thank you for the privilege of reviewing your work. Below you will find my comments, instructions from the Spectrum editorial office, and the reviewer comments.

Revision Guidelines

Sincerely,
Tulip Jhaveri
Editor
Microbiology Spectrum

Reviewer #1 (Comments for the Author):

A. baumannii is one of the important drug-resistant bacteria in current clinical practice, especially the pulmonary infections caused by multidrug-resistant strains (MDRAB), which pose significant challenges to clinical treatment. This article investigates the efficacy of sulbactam combined treatment regimen in monobacterial and polymicrobial pulmonary infections, which holds important clinical significance. The study design is reasonable: it is a multi-center retrospective study that encompasses data

from multiple hospitals, with a large sample size (366 patients), which can better reflect the clinical situation in the real world. Additionally, the study analyzes both microbiological and clinical outcomes, assessing the effectiveness of the treatment regimen from multiple perspectives.

Due to being retrospective studies, there may be selection bias and confounding factors. For example, the treatment regimen, drug dosage, and treatment duration for different patients may be influenced by various factors, making it difficult to completely eliminate the interference of these factors on the research results. It is recommended to further elaborate on the possible impact of these limitations on the research results in the discussion section, and propose the necessity of conducting prospective studies in the future.

The article mentions that the clinical outcome analysis only included patients who received at least 7 days of sulbactam treatment, which may result in the loss of clinical data for some patients. Although this design helps ensure the integrity and reliability of the data, it may also introduce selection bias. It is recommended to supplement the results section with information on the basic characteristics of patients excluded due to insufficient treatment duration, as well as whether there are significant differences between these patients and those included in the analysis.

How to determine whether a patient with positive *Acinetobacter baumannii* culture is infected, especially in cases where multiple microorganisms are simultaneously positive, requires a detailed description.

Reviewer #2 (Comments for the Author):

This retrospective multicenter study evaluates the efficacy of sulbactam in treating both monomicrobial and polymicrobial *A. baumannii* infections in patients with MDR-*A. baumannii* pneumonia. The 2016 Infectious Diseases Society of America and the American Thoracic Society Clinical Practice Guidelines was applied to the pneumonia inclusion criteria, and *A. baumannii* isolates non-susceptible to at least one agent in three or more antimicrobial categories were considered as multidrug resistant (MDR). All patients were treated with sulbactam with or without another anti-*A. baumannii* agent. The 7-day microbial efficacy and 14-day clinical efficacy were evaluated.

Some comments to improve this study are:

- It is well-known that automated systems may not provide accurate susceptibility results for *Acinetobacter baumannii*. It is important to confirm the susceptibility against sulbactam, polymyxin, and tigecycline by a reference method.
- The study evaluated regimens containing sulbactam, demonstrating that combination therapy with polymyxin showed superiority, but is not clear if the sulbactam is enhancing the polymyxin regimen or not. A controlled group treat with polymyxin alone regimen is necessary to evaluate the sulbactam role in treating patients with *A. baumannii* pneumonia.

Response to Reviewers

Dear Reviewers,

Thanks very much to all reviewers for their valuable comments and suggestions on our manuscript Spectrum03355-24. According to the referee's report, we have revised our manuscript carefully. The revised version has been submitted to your journal through the submission system.

The reviewers' questions are answered as follows:

Reviewer 1

Comment 1: Due to being retrospective studies, there may be selection bias and confounding factors. For example, the treatment regimen, drug dosage, and treatment duration for different patients may be influenced by various factors, making it difficult to completely eliminate the interference of these factors on the research results. It is recommended to further elaborate on the possible impact of these limitations on the research results in the discussion section, and propose the necessity of conducting prospective studies in the future.

Reply: Before the start of this study, we had fully recognized that confounding factors are one of the important sources of bias in retrospective observational studies. Controlling confounding bias is particularly crucial in observational comparative studies of efficacy^[1]. To correct the confounding factors in the study, we employed the multivariate logistic regression analysis method in statistical analysis^[2, 3]. First, through univariate analysis, we screened potential confounding factors from the baseline characteristics, such as the course of antibacterial drug treatment. These potential confounding factors are variables that are related to both the intervention factor and the study outcome, but are not intermediate variables in the causal pathway between exposure and outcome. Then, we selected these factors and entered them to the multivariate model analysis together with the efficacy outcome variables. The performance and suitability of the multivariate model were evaluated through the Hosmer-Lemeshow goodness-of-fit test, thereby enhancing the reliability of the research findings.

Nevertheless, some unidentifiable and immeasurable confounding factors remain inevitable. Therefore, we elaborated the limitations of the study in the discussion section and reiterated that the type, quantity, and duration of antibiotic treatments varied among groups, which may introduce certain biases into the results. We also underscore the necessity of conducting prospective studies in the future.

The relevant references are as follows:

[1] Brookhart MA, Stürmer T, Glynn RJ, Rassen J, Schneeweiss S. 2010. Confounding control in healthcare database research: challenges and potential approaches. *Med Care* 48(6 Suppl): S114-S120. <https://doi.org/10.1097/MLR.0b013e3181d8bebe3>.

[2] Klungel OH, Martens EP, Psaty BM, Grobbee DE, Sullivan SD, Stricker BH, Leufkens HG, de Boer A. 2004. Methods to assess intended effects of drug treatment in observational studies are reviewed. *J Clin Epidemiol* 57:1223-1231. <https://doi.org/10.1016/j.jclinepi.2004.03.011>.

[3] Wunsch H, Linde-Zwirble WT, Angus DC. 2006. Methods to adjust for bias and confounding in critical care health services research involving observational data. *J Crit Care* 21:1-7.

Comment 2: The article mentions that the clinical outcome analysis only included patients who received at least 7 days of sulbactam treatment, which may result in the loss of clinical data for some patients. Although this design helps ensure the integrity and reliability of the data, it may also introduce selection bias. It is recommended to supplement the results section with information on the basic characteristics of patients excluded due to insufficient treatment duration, as well as whether there are significant differences between these patients and those included in the analysis.

Reply: As noted, excluding patients who received sulbactam for less than 7 days may introduce selection bias and result in the loss of certain clinical data. In our analysis of 14-day clinical efficacy, we carefully evaluated whether to include these cases or not. Given the presence of other antimicrobial agents in combination therapy, patients receiving sulbactam for less than 7 days may not accurately represent the effects of sulbactam.

To support this decision, we consulted several relevant studies on antimicrobial efficacy. For example, Zheng et al. ^[1] analyzed the effects of colistin on multidrug-resistant *Acinetobacter baumannii* (MDRAB)-related pneumonia in the *Journal of Microbiology, Immunology and Infection*, and established an inclusion criterion of at least 7 days of colistin treatment (monotherapy or combination therapy) to evaluate clinical outcomes. Similarly, Sirijatuphat et al. ^[2] assessed the efficacy of colistin monotherapy and colistin-fosfomicin combination therapy for carbapenem-resistant *A. baumannii* (CRAB) infections in *Antimicrobial Agents and Chemotherapy*, and defined a treatment duration of 7 to 14 days. Zalts et al. ^[3] retrospectively studied intravenous colistin and ampicillin-sulbactam for CRAB ventilator-associated pneumonia (VAP) and set a treatment duration of 7 to 10 days, with extensions as needed.

Furthermore, patients treated with sulbactam for less than 7 days often lacked complete objective data, such as laboratory results and imaging studies. This incompleteness could compromise the reliability of clinical efficacy analyses, as subjective judgments might introduce uncertainty. To ensure the integrity and reliability of our findings and in alignment with the criteria established in the literature, our 14-day clinical efficacy analysis excluded patients with sulbactam treatment duration of less than 7 days.

To address potential concerns from reviewers, we compared the baseline characteristics of patients treated with sulbactam for ≥ 3 - <7 days against those treated for ≥ 7 days. Results revealed significant differences between the two groups in terms of gender, mechanical ventilation, ICU admission, duration of antimicrobial therapy, comorbid cerebrovascular diseases, comorbid *Staphylococcus* infections, and sulbactam combination regimens. These differences suggest that patients excluded from the analysis deviated from the overall characteristics of the included cohort. Consequently, we excluded the clinical data from patients with sulbactam treatment duration less than 7 days.

For transparency, the baseline characteristics and comparisons of these excluded

patients are provided in Supplementary Table 3.

The relevant references are as follows:

[1] Zheng JY, Huang SS, Huang SH, Ye JJ. 2020. Colistin for pneumonia involving multidrug-resistant *Acinetobacter calcoaceticus*-*Acinetobacter baumannii* complex. *J Microbiol Immunol Infect* 53:854-865. <https://doi.org/10.1016/j.jmii.2019.08.007>.

[2] Sirijatuphat R, Thamlikitkul V. 2014. Preliminary study of colistin versus colistin plus fosfomycin for treatment of carbapenem-resistant *Acinetobacter baumannii* infections. *Antimicrob Agents Chemother* 58:5598-5601. <https://doi.org/10.1128/AAC.02435-13>.

[3] Zalts R, Neuberger A, Hussein K, Raz-Pasteur A, Geffen Y, Mashiach T, Finkelstein R. 2016. Treatment of Carbapenem-Resistant *Acinetobacter baumannii* Ventilator-Associated Pneumonia: Retrospective Comparison Between Intravenous Colistin and Intravenous Ampicillin-Sulbactam. *Am J Ther* 23:e78-e85. <https://doi.org/10.1097/MJT.0b013e3182a32df3>.

TABLE S3 Characteristics of patients with sulbactam treatment ≥ 3 - <7 d and ≥ 7 d in the 14-day clinical efficacy analysis

Characteristic	Sulbactam treatment	Sulbactam treatment	p Value ^a
	≥ 3 - <7 days (n=85)	≥ 7 days (n=281)	
Age (year, median [IQR])	69[55, 79.5]	67[56, 74]	0.150
Male	68(80.0)	177(63.0)	0.004
APACHE II score (median [IQR])	19[14, 23]	19[14.5, 22]	0.998
Mechanical Ventilation	62(72.9)	233(82.9)	0.044
Admit to ICU	76(89.4)	270(96.1)	0.036
Treatment course (day, median [IQR])	5[4, 6]	12[9, 17]	<0.001
Chronic underlying diseases	59(69.4)	206(73.3)	0.491
Diabetes	18(21.2)	63(22.4)	0.882
Hypertension	41(48.2)	142(50.5)	0.805
Heart disease	21(24.7)	76(27.0)	0.678
Cerebrovascular disease	23(27.1)	38(13.5)	0.005
Malignant tumor	3(3.5)	25(8.9)	0.110
Complicated with infection other than pneumonia	31(36.5)	121(43.1)	0.316
Bloodstream infection	10(11.8)	57(20.3)	0.080
Urinary tract infection	9(10.6)	43(15.3)	0.296
Intracranial infection	7(8.2)	8(2.8)	0.060
Skin and soft tissue infection	2(2.4)	15(5.3)	0.394
Combined with other pathogens	64(75.3)	233(82.9)	0.153
Klebsiella pneumoniae	25(29.4)	90(32.0)	0.691
Pseudomonas aeruginosa	21(24.7)	101(35.9)	0.066
Stenotrophomonas maltophilia	10(11.8)	53(18.9)	0.143

Staphylococcus species	6(7.1)	46(16.4)	0.033
Escherichia coli	3(3.5)	17(6.0)	0.533
Sulbactam daily dosage			
≤4g (ref)	69(81.2)	192(68.3)	0.069
≥6g- < 8g	9(10.6)	46(16.4)	
≥8g	7(8.2)	43(15.3)	
Sulbactam combination regimens			
Sulbactam-based (ref)	43(50.6)	92(32.7)	0.012
Sulbactam + tigecycline	28(32.9)	109(38.8)	
Sulbactam + polymyxin	8(9.4)	32(11.4)	
Sulbactam + polymyxin + tigecycline	6(7.1)	48(17.1)	

Data are presented as median (25th-75th percentiles) or N (%). Statistically significant p values are highlighted in bold. APACHE, Acute Physiology and Chronic Health Evaluation; ICU, intensive care unit.

^ap Values were calculated by Mann-Whitney U test, or chi-square test as appropriate.

Comment 3: How to determine whether a patient with positive *Acinetobacter baumannii* culture is infected, especially in cases where multiple microorganisms are simultaneously positive, requires a detailed description.

Reply: In the methodology section, we have provided the definition of MDRAB-related pneumonia and supplemented it with relevant literature ^[1]. To enhance the clarity for reviewers regarding whether patients with positive cultures for *A. baumannii* are indeed infected, we have elaborated the definitions of monomicrobial *A. baumannii* (mono-AB) and polymicrobial *A. baumannii* (poly-AB). Mono-AB is defined as only positive culture results for *A. baumannii* with time-related symptoms between three days before sample collection and the evaluation day ^[1]. Poly-AB is defined as an infection with positive culture of *A. baumannii* with time-related symptoms, and positive culture results of other bacteria, regardless of infection or colonization ^[2, 3]. Notably, we determined *A. baumannii* to be the pathogenically responsible pathogen in polymicrobial *A. baumannii*-associated pneumonia, but did not differentiate the other bacteria with positive culture results in terms of infection or colonization.

The relevant references are as follows:

[1] Zheng JY, Huang SS, Huang SH, Ye JJ. 2020. Colistin for pneumonia involving multidrug-resistant *Acinetobacter calcoaceticus*-*Acinetobacter baumannii* complex. *J Microbiol Immunol Infect* 53:854-865. <https://doi.org/10.1016/j.jmii.2019.08.007>.

[2] Appaneal HJ, Lopes VV, LaPlante KL, Caffrey AR. 2022. Treatment, Clinical Outcomes, and Predictors of Mortality among a National Cohort of Admitted Patients with *Acinetobacter baumannii* Infection. *Antimicrob Agents Chemother* 66:e0197521. <https://doi.org/10.1128/AAC.01975-21>.

[3] Hardak E, Avivi I, Berkun L, Raz-Pasteur A, Lavi N, Geffen Y, Yigla M, Oren I. 2016.

Polymicrobial pulmonary infection in patients with hematological malignancies: prevalence, co-pathogens, course and outcome. *Infection* 44:491–497. <https://doi.org/10.1007/s15010-016-0873-3>.

Reviewer 2

Comment 1: It is well-known that automated systems may not provide accurate susceptibility results for *Acinetobacter baumannii*. It is important to confirm the susceptibility against sulbactam, polymyxin, and tigecycline by a reference method.

Reply: We have supplemented the description of the methods for determining the drug sensitivity results of sulbactam, polymyxin, and tigecycline according to your suggestions. Please refer to the methodology section for details. The microbiology laboratories of the hospitals determined the minimum inhibitory concentrations (MICs) of the above three antimicrobial agents using the Agar dilution method. The drug sensitivity breakpoints of polymyxin are determined according to the 2020 standards of the European Committee on Antimicrobial Susceptibility Testing (EUCAST). The determination basis of cefoperazone/sulbactam is the breakpoint standard of cefoperazone, and the situation of sulbactam is determined according to the drug sensitivity results of cefoperazone/sulbactam. The drug sensitivity breakpoints of tigecycline are cited from the standards of the U.S. Food and Drug Administration (FDA) ^[1].

The supplementary references are as follows:

[1] United States Food and Drug Administration. Accessed 24 December 2023. Antibacterial Susceptibility Test Interpretive Criteria. <https://www.fda.gov/drugs/development-resources/antibacterial-susceptibility-test-interpretive-criteria>.

Comment 2: The study evaluated regimens containing sulbactam, demonstrating that combination therapy with polymyxin showed superiority, but is not clear if the sulbactam is enhancing the polymyxin regimen or not. A controlled group treat with polymyxin alone regimen is necessary to evaluate the sulbactam role in treating patients with *A. baumannii* pneumonia.

Reply: In recent years, relevant studies confirm that sulbactam can significantly enhance the therapeutic efficacy of polymyxins against *A. baumannii* infections. For instance, Srisakul et al. ^[1] tested colistin-resistant *A. baumannii* and demonstrated a synergy rate of 86.7% when sulbactam was combined with colistin, which is notably higher than the efficacy of sulbactam (resistance rate of 85.89%) or colistin (15.14%) used alone. Furthermore, Qu et al. ^[2] reported that the combination of sulbactam and polymyxin B exhibited the highest synergistic effect (82.35%). For carbapenem-resistant *A. baumannii* (CRAB), sulbactam-based combinations showed greater synergy than the combination of polymyxin B and tigecycline. A systematic review and network meta-analysis involving 2,529 patients ^[3] found that the combination of sulbactam and colistin had superior microbiological efficacy compared to monotherapy with colistin and the combination of colistin and

tigecycline in treating multidrug-resistant *A. baumannii* (MDRAB) and pan-resistant *A. baumannii* infections. These findings indicate that the combination of sulbactam and polymyxins exhibits a favorable synergistic effect and clinical efficacy against *A. baumannii* compared to monotherapy with polymyxins.

The sample size utilizing polymyxin treatment in this study is limited, as only 39 cases involved the combination of sulbactam and polymyxin. Furthermore, cases treated solely with polymyxin were not included in the initial design of this study. Given the current limited sample size, conducting a retrospective analysis is not appropriate. Additionally, due to the heterogeneous resistance associated with polymyxins, monotherapy with polymyxins is generally not recommended [4, 5]. Nevertheless, in accordance with your suggestion, we may consider doing prospective research in this area in the future.

The relevant references are as follows:

- [1] Srisakul S, Wannigama DL, Higgins PG, Hurst C, Abe S, Hongsing P, Saethang T, Luk-In S, Liao T, Kueakulpattana N, Shein AMS, Gan L, Kupwiwat R, Tanasatitchai C, Wapeesittipan P, Phattharapornjaroen P, Badavath VN, Leelahavanichkul A, Chatsuwan T. 2022. Overcoming addition of phosphoethanolamine to lipid A mediated colistin resistance in *Acinetobacter baumannii* clinical isolates with colistin-sulbactam combination therapy. *Sci Rep* 12:11390. <https://doi.org/10.1038/s41598-022-15386-1>.
- [2] Qu J, Yu R, Wang Q, Feng C, Lv X. 2020. Synergistic Antibacterial Activity of Combined Antimicrobials and the Clinical Outcome of Patients With Carbapenemase-Producing *Acinetobacter baumannii* Infection. *Front Microbiol* 11:541423. <https://doi.org/10.3389/fmicb.2020.541423>.
- [3] Kengkla K, Kongpakwattana K, Saokaew S, Apisarnthanarak A, Chaiyakunapruk N. 2018. Comparative efficacy and safety of treatment options for MDR and XDR *Acinetobacter baumannii* infections: a systematic review and network meta-analysis. *J Antimicrob Chemother* 73:22-32. <https://doi.org/10.1093/jac/dkx368>.
- [4] Cai Y, Chai D, Wang R, Liang B, Bai N. 2012. Colistin resistance of *Acinetobacter baumannii*: clinical reports, mechanisms and antimicrobial strategies. *J Antimicrob Chemother* 67:1607-1615. <https://doi.org/10.1093/jac/dks084>.
- [5] Tsuji BT, Pogue JM, Zavascki AP, Paul M, Daikos GL, Forrest A, Giacobbe DR, Viscoli C, Giamarellou H, Karaiskos I, Kaye D, Mouton JW, Tam VH, Thamlikitkul V, Wunderink RG, Li J, Nation RL, Kaye KS. 2019. International Consensus Guidelines for the Optimal Use of the Polymyxins: Endorsed by the American College of Clinical Pharmacy (ACCP), European Society of Clinical Microbiology and Infectious Diseases (ESCMID), Infectious Diseases Society of America (IDSA), International Society for Anti-infective Pharmacology (ISAP), Society of Critical Care Medicine (SCCM), and Society of Infectious Diseases Pharmacists (SIDP). *Pharmacotherapy* 39:10-39. <https://doi.org/10.1002/phar.2209>.

We are extremely grateful to the reviewers for their insightful feedback, which is highly valuable for improving the quality of our work. We hope that this revised manuscript has been satisfactorily improved and will be accepted for publication in your esteemed journal.

Re: Spectrum03355-24R1 (Evaluation of the Efficacy of Sulbactam Combination Therapy for Monomicrobial and Polymicrobial Pulmonary Infections Caused by Multidrug-Resistant *Acinetobacter baumannii*)

Dear Dr. wenkui sun:

Your manuscript has been accepted, and I am forwarding it to the ASM production staff for publication. Your paper will first be checked to make sure all elements meet the technical requirements. ASM staff will contact you if anything needs to be revised before copyediting and production can begin. Otherwise, you will be notified when your proofs are ready to be viewed.

Sincerely,
Tulip Jhaveri
Editor
Microbiology Spectrum

Reviewer #1 (Comments for the Author):

The author has answered my questions regarding several issues in the response and made revisions to the manuscript. I agree to accept the current version for publication.